# Potassium Dehydroandrograpolide Succinate Targets NRP1 Mediated VEGFR2/VE-Cadherin Signaling Pathway to Promote Endothelial Barrier Repair

**DOI:** 10.3390/ijms24043096

**Published:** 2023-02-04

**Authors:** Zheng Wang, Xiao Wu, Jiali Li, Qiru Guo, Zhong Jin, Hongfei Li, Bing Liang, Wangming Hu, Huan Xu, Liangqin Shi, Lan Yang, Yong Wang

**Affiliations:** College of Basic Medicine, Chengdu University of Traditional Chinese Medicine, Chengdu 610000, China

**Keywords:** PDA, NRP1, VEGF signaling pathway, VE-Cad, endothelial barrier, pathological vascular remodeling

## Abstract

Impairment of vascular endothelial integrity is associated with various vascular diseases. Our previous studies demonstrated that andrographolide is critical to maintaining gastric vascular homeostasis, as well as to regulating pathological vascular remodeling. Potassium dehydroandrograpolide succinate (PDA), a derivative of andrographolide, has been clinically used for the therapeutic treatment of inflammatory diseases. This study aimed to determine whether PDA promotes endothelial barrier repair in pathological vascular remodeling. Partial ligation of the carotid artery in ApoE^−/−^ mice was used to evaluate whether PDA can regulate pathological vascular remodeling. A flow cytometry assay, BRDU incorporation assay, Boyden chamber cell migration assay, spheroid sprouting assay and Matrigel-based tube formation assay were performed to determine whether PDA can regulate the proliferation and motility of HUVEC. A molecular docking simulation and CO-immunoprecipitation assay were performed to observe protein interactions. We observed that PDA induced pathological vascular remodeling characterized by enhanced neointima formation. PDA treatment significantly enhanced the proliferation and migration of vascular endothelial cells. Investigating the potential mechanisms and signaling pathways, we observed that PDA induced endothelial NRP1 expression and activated the VEGF signaling pathway. Knockdown of NRP1 using siRNA transfection attenuated PDA-induced VEGFR2 expression. The interaction between NRP1 and VEGFR2 caused VE-Cad-dependent endothelial barrier impairment, which was characterized by enhanced vascular inflammation. Our study demonstrated that PDA plays a critical role in promoting endothelial barrier repair in pathological vascular remodeling.

## 1. Introduction

The integrity of the endothelial barrier, which is composed of a monolayer of endothelial cells, is pivotal for the regulation of the communication between circulating blood and tissues through delivery of molecules and circulating substances across the endothelial barrier [1]. Impairment of vascular endothelial integrity causes vascular permeability and leukocyte extravasation [2], which is of pivotal importance in the development of several pathological respiratory diseases, including inflammation and interstitial oedema of the lung, sepsis and acute respiratory distress syndrome [3,4,5,6], and vascular pathological diseases, such as ischemic stroke [3], atherosclerosis [7], hypertension [8], retinal vascular leakage and neovascularization [9]. Dysfunction in the endothelial barrier promotes cancer cell extravasation [10].

Interendothelial junctions play an important role in regulating endothelial function, such as vasculogenesis, angiogenesis and vascular permeability [11]. Classical cadherins are the principle adhesive proteins at cohesive intercellular junctions and are essential molecules for morphogenesis and tissue homeostasis [12]. Cadherin endocytosis governs the plasticity of cell contacts and regulates cell migration and angiogenesis [13]. Vascular endothelial cadherin (VE-Cad) is a strictly endothelium-specific adhesion molecule located at junctions between endothelial cells [14]. VE-Cad-regulated adhesion plays a critical role in controlling vascular permeability and leukocyte extravasation. VE-Cad is essential for the formation of new vessel networks due to its important roles in regulating cell proliferation and apoptosis and modulating vascular endothelial growth factor receptor functions through modulation of endothelial cell VEGF and FGF signaling pathways [14,15]. VE-Cad is also required for vascular integrity and normal organ functions [16]. Disturbed hemodynamic forces [3], inflammatory diseases [17] and metabolic disorders can phosphorylate VE-cadherin, which contributes to the regulation of vascular permeability.

Neuropilin 1 (NRP1), a transmembrane glycoprotein, is required for embryonic neuronal and vascular development [18]. The endothelial cell NRP1 binds vascular endothelial growth factor (VEGF) A and forms heterocomplexes with VEGFR2 to enhance the VEGF signaling pathway, which is essential for both developmental angiogenesis and pathological angiogenesis [18,19,20,21]. NRP1 is a key regulator of VEGF-dependent arteriogenesis and arterial morphogenesis in the developing heart, kidney, hindlimb and eye [22]. NRP1 promotes endothelial cell proliferation, motility and capillary-like tube formation, whereas it inhibits apoptosis [23]. The endothelial tip cell NRP1 promotes sprouting of angiogenetic vessels [24]. Interaction between reelin and NRP1 also regulates neocortical dendrite development [25].

Our previous studies demonstrated that andrographolide is critical to maintaining gastric vascular homeostasis, as well as to regulating pathological vascular remodeling [26,27]. Potassium dehydroandrograpolide succinate (PDA), a derivative of andrographolide, has been clinically used for therapeutic treatment of inflammatory diseases [28]. However, as a derivative synthesized from andrographolide, it is still largely unknown whether PDA has potential medicinal value for the treatment of cardiovascular diseases. This study aimed to determine the molecular mechanism whereby PDA repairs damaged intimal barriers during pathological vascular remodeling.

## 2. Results

### 2.1. PDA Promotes Partial Carotid Artery Ligation-Induced Neointima Formation

Partial carotid artery ligation is an acute disturbed blood flow model leading to rapid endothelial dysfunction [29]. ApoE knockout mice were fed a Western diet and subjected to consecutive administration of PDA via intraperitoneal injection for 5 days. Partial carotid artery ligation was performed and arteries were harvested at 7 and 28 days after surgery (Figure 1A). Morphological change was determined by H&E staining. We observed that PDA treatment for 7 days did not change the area of the medium smooth muscle layer (Appendix A), whereas the neointima area, as well as the ratio of the neointima area to the medium smooth muscle layer area, was dramatically increased (Figure 1B–D). Morphological change was also determined after PDA treatment for 28 days. PDA treatment resulted in a conspicuous enhancement of the neointima area and the ratio of the neointima area to the medium smooth muscle layer area (Figure 1E–G). However, PDA treatment did not change the area of the medium smooth muscle layer (Appendix A). These data indicate that PDA promotes partial carotid artery ligation-induced neointima formation.

### 2.2. PDA Promotes Vascular Endothelial Cell Proliferation

Dysfunction in the endothelial barrier contributes to pathological vascular remodeling [30]. We sought to determine whether PDA can regulate endothelial cell proliferation. Different doses of PDA treatment did not affect HUVEC-C viability as evaluated by the CCK8 assay after 1 μM, 5 μM, 10 μM and 20 μM treatments (Appendix A). However, PDA treatment promoted HUVEC-C growth. After PDA treatment, the cell numbers were counted at different time points. We observed that HUVEC numbers were significant increased after PDA treatment for 24 h, 48 h and 72 h (Figure 2A). After PDA treatment, the cell cycle was determined using flow cytometry analysis. The numbers of cells that entered the S phase were obviously increased (Figure 2B; Appendix A). The BRDU incorporation assay was also performed, and our results indicated that PDA treatment significantly increased BRDU-incorporated HUVEC-C (Figure 2C,D). We next performed real-time PCR to evaluate the transcription levels of several genes critical in regulating cell cycles. Our data exhibited that the transcription levels of P14arf, P15ink4b, P18ink4c, P19arf and P27kip1 were suppressed, whereas the transcription levels of CyclinD1 and PCNA were enhanced (Figure 2E). We also performed Western blotting to evaluate expression levels of the cell growth marker gene, and our results showed that PDA treatment for 3 h, 6 h, 12 h, 24 h and 48 h dramatically induced PCNA expression (Figure 2F,G). Our IHC staining, performed against PCNA antibody on slides from partially ligated carotid artery, indicated that PCNA expression was significant induced after PDA treatment for 7 days (Figure 2H,I). The capillary endothelial cell is a promising model for the study of endothelial cell activity. We used a cardiotoxin-induced skeletal muscle injury model to determine whether PDA regulates endothelial cell proliferation during angiogenesis [31]. We performed IHC staining against the CD31 antibody and observed that PDA treatment significant enhanced vascular vessel numbers during skeletal muscle regeneration (Appendix A). We also performed IF staining against the PCNA antibody and observed that more PCNA-positive endothelial cells were exhibited during angiogenesis following PDA treatment (Figure 2J,K). These data indicate that PDA promotes vascular endothelial cell proliferation.

### 2.3. PDA Promotes Vascular Endothelial Cells Migration

We next sought to determine whether PDA promotes vascular endothelial cell migration. The extracellular matrix plays a pivotal role in regulating cell migration [32]. We observed that transcription levels of integrin, versican 0, versican 1 and VE-Cad were obviously increased following PDA treatment, as evaluated by real-time PCR (Figure 3A). We next performed the Boyden chamber migration assay and observed that PDA treatment increased the number of migrated HUVEC-C (Figure 3B; Appendix A). PDA treatment promoted HUVEC-C migration, as also exhibited in our spheroid sprouting assay, which was characterized by significant inducing sprouting numbers and sprouting lengths (Figure 3C–E). We further performed a Matrigel-based tube formation assay to evaluate whether PDA suppresses angiogenesis in vitro. Migrating HUVEC-Cs touched each other and tubes were formed. PDA promoted HUVEC-C migration and more tube-like phenotypes, including mesh areas and partially tube-like branches, were exhibited after PDA treatment (Figure 3F–H). Our data indicate that PDA promotes vascular endothelial cell migration.

### 2.4. PDA Induces Vascular Endothelial Cell NRP1 Expression after Vascular Injury

To study the underlying mechanism whereby PDA promotes endothelial barrier repair, transcription levels of multiple targets were examined using real-time PCR, and we observed that NRP1, VE-Cad and VEGF signaling pathways were prominently induced following PDA treatment (Appendix A). We first sought to determine whether PDA induces vascular endothelial cell NRP1 expression. We treated HUVEC-Cs with different doses of PDA, including 1 μΜ, 5 μΜ, 10 μΜ, 15 μΜ and 20 μΜ, and Western blotting was performed to evaluate NRP1 protein levels. Our results indicated that 1 μΜ, 5 μΜ and 10 μΜ of PDA treatment dramatically induced HUVEC-C NRP1 protein levels (Figure 4A,B). Proteins from HUVEC-Cs were also harvested at different time points, including 3 h, 6 h, 12 h, 24 h and 48 h, following 10 μΜ of PDA treatment. Our results indicated that 6 h, 12 h and 24 h of PDA treatment significantly induced NRP1 expression (Figure 4C,D). Our IF staining against NRP1 antibody also validated that PDA treatment promoted NRP1 expression in HUVECs (Figure 4E,F). We further evaluated NRP1 expression in our animal studies. Our IHC and IF staining against the NRP1 antibody on a slide using partially ligated carotid artery data indicated that NRP1 expression was significantly induced after PDA treatment for 7 days, and enhanced expression of NRP1 was exhibited in the neointima area and endothelial cells, with dramatically enhanced expression of NRP1 located in the endothelial layer (Figure 4G,H; Appendix A). A skeletal muscle injury-induced angiogenesis model was used to evaluate endothelial NRP1 expression during angiogenesis. Our IF staining against NRP1 antibody indicated that PDA dramatically induced capillary endothelial NRP1 expression (Figure 4I,J). These data demonstrate that PDA induces vascular endothelial cell NRP1 expression during pathological vascular remodeling.

### 2.5. PDA Activates VEGF Signaling Pathway after Vascular Injury

We next sought to determine whether PDA can activate the VEGF signaling pathway. We treated HUVEC-Cs with PDA and real-time PCR was performed to evaluate VEGF signaling targets. Our results indicated that VEGF121, VEGF165, VEGF189 and VEGFR2 were significant induced (Figure 5A). The VEGFR2 protein level was also determined using Western blotting, and we observed that 1 μΜ, 5 μΜ, 10 μΜ and 15 μΜ of PDA significantly promoted VEGFR2 expression in HUVECs (Figure 5B,C). Our IF staining against VEGFR2 antibody also confirmed that PDA treatment obviously induced VEGFR expression in HUVECs (Figure 5D,E). We next sought to determine whether PDA induces VEGFR2 expression in animal models. In a carotid partial ligation model, our IF and IHC staining against VEGFR2 antibody indicated that PDA promoted carotid artery endothelial cell VEGFR2 expression after carotid artery ligation for 7 days (Figure 5F,G; Appendix A). A skeletal muscle injury-induced angiogenesis model was also used, and we observed that PDA promoted endothelial cell VEGFR2 expression during angiogenesis (Figure 5H,I). Our data indicate that PDA activates the VEGF signaling pathway after vascular injury.

### 2.6. PDA Causes VE-Cad-Dependent Vascular Endothelial Barrier Repair

VE-Cad mediation of endothelial barriers is critical to maintaining vascular homeostasis. Our real-time PCR data indicated that PDA treatment enhanced VE-Cad transcription levels in HUVEC-Cs (Figure 6A). PDA treatment also increased VE-Cad protein levels. We treated HUVEC-Cs with different doses of PDA, including 1 μM, 5 μM, 10 μM, 15 μM and 20 μM, and Western blotting data indicated that 1 μM, 5 μM and 10 μM of PDA obviously induced VE-Cad protein levels (Figure 6B,C). Our IF staining against the VE-Cad antibody indicated that the majority of VE-Cad was expressed in the cell membrane, and PDA treatment dramatically increased cell membrane VE-Cad expression within HUVECs (Figure 6D,E). The integrity of the vascular endothelial barrier determines the homeostasis of blood vessels. VE-Cad is critical to maintaining and controlling endothelial contacts, which is important for control of vascular permeability and leukocyte extravasation [14]. Whether activation of VE-Cad by PDA treatment can regulate the vascular endothelial barrier is largely unknown. We treated HUVEC-Cs with PDA and real-time PCR was performed to determine transcription levels of inflammation-related genes. Our real-time PCR data indicated that PDA treatment enhanced inflammation-related gene transcription in HUVEC-Cs, including IL-6, ICAM1, MCP1 and E-selectin (Figure 6F). The attachment of THP-1 monocytes to HUVECs after PDA treatment was also evaluated. Our results indicated that PDA enhanced the cell numbers where THP-1 monocytes were attached to HUVECs (Figure 6G,H). We next sought to determine whether PDA induces inflammatory infiltration in animal models. In a carotid partial ligation model, our IF staining against the MAC-2 antibody indicated that PDA increased vascular macrophage numbers (Figure 6I,J). These data demonstrate that PDA treatment causes VE-Cad-dependent vascular endothelial barrier repair.

### 2.7. PDA Regulates the Interaction between NRP1, VEGFR and VE-Cad within Vascular Endothelial Cells

NRP1, a co-receptor for VEGF-A, can form heterocomplexes with VEGFR2 to activate the VEGF signaling pathway. However, the relationship between NRP1, VEGFR2 and VE-Cad is not well-defined. We first sought to analyze a molecular docking simulation of PDA with NRP1, VE-Cad and VEGFR2, determining the binding energy using Autodock Vina 1.5.6 software. The three-dimensional structures of NRP1 and VEGFR2 were obtained from the RCSBPDB database (http://www.rcsb.org/, accessed on 16 August 2022) and that of VE-Cad from the SWISS-Model database (http://swissmodel.expasy.org/, accessed on 20 August 2022). Multiple potential binding motifs between PDA and NRP1, VE-Cad and VEGFR2 were exhibited (Figure 7A). When values for the binding energy were less than zero, those proteins were considered to spontaneously bind and interact with each other. Binding energy values of −9.1 were exhibited between PDA and NRP1, −3.5 between PDA and VE-Cad and −7.4 between PDA and VEGFR2 (Figure 7B). These data indicate that PDA potentially interacts with NRP1, VE-Cad and VEGFR2. We next sought to determine the expression pattern between NRP1, VEGFR2 and VE-Cad. In a skeletal muscle injury-induced angiogenesis model, our IF staining against NRP1 and VEGFR2 antibodies indicated that NRP1 and VEGFR2 were co-localized within vascular endothelial cells during angiogenesis (Figure 7C). Co-localization of NRP1 and VEGFR2 was confirmed in HUVECs (Appendix A). We also observed that NRP1 and VE-Cad were co-localized in HUVECs (Appendix A). We further evaluated the interaction between NRP1, VE-Cad and VEGFR2 in endothelial cells using a CO-immunoprecipitation assay. Pooled protein was prepared form HUVECs. After immunoprecipitation, Western blotting was used to visualize interaction targets. Our data indicated that both NRP1 and VE-Cad can interact with each other, and VEGFR2 can interact with NRP1 and VE-Cad (Figure 7E). These data indicate that PDA can regulate interactions between NRP1, VEGFR and VE-Cad within vascular endothelial cells.

### 2.8. PDA Regulates Vascular Endothelial Barrier Function through the NRP1/VEGFR2/VE-Cad Signaling Pathway

To determine whether PDA regulates the endothelial barrier through activation of the NRP1/VEGFR2/VE-Cad signaling pathway, we knocked down NRP1 in HUVECs using small interfering RNA (siRNA) that specifically targeted NRP1. We first performed real-time PCR to validate the NRP1 knockdown efficiency. Different small interfering RNA sequences were designed. The S1 small interfering RNA sequence was used for further studies, the knockdown efficiency of which was confirmed in HUVECs (Appendix A) and HUVEC-Cs (Appendix A). PDA treatment dramatically induced NRP1, VEGFR2 and VE-Cad expression. However, the relationship between NRP1, VEGFR2 and VE-Cad is not well-defined. In order to determine whether NRP1 regulates VEGFR2 and VE-Cad expression, after knockdown of NRP1 in HUVECs, we performed real-time PCR to evaluate transcription levels of VEGF signaling pathway targets and VE-Cad, and our results indicated that expression of VEGFR2 and VE-Cad dramatically decreased (Figure 8A). Knockdown of NRP1 attenuated PDA treatment-induced VEGFR2 and VE-Cad transcription in HUVECs (Figure 8B). Knockdown of NRP1 also attenuated PDA treatment-induced VE-Cad expression (Figure 8C,D). These data indicate that PDA regulated the NRP1-mediated VEGF/VA-Cad signaling pathway. We sought to determine whether PDA interrupted the vascular endothelial barrier through the NRP1/VEGFR2/VE-Cad signaling pathway, and we knocked down NRP1 in HUVECs following PDA treatment and performed real-time PCR to evaluate transcription levels of inflammation-related genes. Our data indicated that knockdown of NRP1 attenuated PDA treatment-induced transcription of IL-6, ICAM1, VCAM1, MCP1 and E-selectin (Figure 8E). The attachment of THP-1 monocytes to HUVECs assay was also performed. Our results indicated that knockdown of NRP1 attenuated THP-1 monocyte attachment following PDA treatment (Figure 8F,G). These data demonstrate that PDA regulates vascular endothelial barrier function through the NRP1/VEGFR2/VE-Cad signaling pathway.

In summary, disturbed blood flow induces vascular endothelial cell NRP1 expression, which is enhanced by PDA treatment, consequently activating the VEGF signaling pathway. The interaction between NRP1and VEGFR2 can enhance VE-Cad recruitment, resulting in the impairment of vascular homeostasis through interruptions to the vascular endothelial barrier function, eventually inducing pathological vascular remodeling (Figure 8H).

## 3. Discussion

This study provides evidence that PDA promotes endothelial barrier repair in pathological vascular remodeling. Disturbed blood flow caused pathological vascular remodeling by damaging the vascular endothelium. Our in vitro study showed that PDA promotes high expression of NRP1, activates the VEGF signaling pathway, accelerates the recruitment of VE-Cad through the interaction of NRP1 with VEGFR2 and, ultimately, promotes the proliferation and migration of umbilical vein endothelial cells, which facilitates the repair of the endothelial barrier and inhibits pathological vascular remodeling. However, our in vivo findings suggest that PDA treatment significantly increases the area of the neointimal area, which is puzzling as well as interesting. Therefore, we can make a hypothesis about this finding. We have already established in in vitro experiments that PDA has the potential to repair the endothelial barrier and that this effect is likely to be bidirectional. Do the pro-proliferative and pro-migratory effects of PDA on endothelial cells then also apply to smooth muscle cells in the vascular mesangial layer? Furthermore, do the large amounts of growth factors and inflammatory mediators released by proliferating endothelial cells also have indirect effects on smooth muscle cells? Perhaps this is due to the significant cell proliferation-promoting ability of PDA in vivo.

### 3.1. PDA Did Not Change HUVEC Viability

PDA, a derivative of andrographolide, has been clinically used for the therapeutic treatment of inflammatory diseases. Our previous studies demonstrated that andrographolide plays a critical role in maintaining vascular homeostasis, as well as in regulating smooth muscle cell phenotypic switch-dependent pathological vascular remodeling [26,27]. However, whether PDA is involved in regulating endothelial barrier repair in pathological vascular remodeling is not known. In a partially ligated murine carotid artery model, we observed that PDA significantly induced neointima formation, and macrophages were exhibited within the vascular neointima area. This indicated that the intima was entering a state of inflammation. However, PDA did not change the viability of either the smooth muscle cells or the endothelial cells, as detected by the CCK8 assay. Our data suggest that disturbed blood flow-induced vascular remodeling is possibly initiated by some other signaling pathways, which contribute to regulating multiple biological processes. The vascular endothelium is known to be a critical tissue that is crucial for the regulation of vascular tone, cell growth, cell motility and interactions between leukocytes, thrombocytes and vascular walls [33].

NRP1 is expressed by neurons, blood vessels, immune cells and many other cell types and binds a range of structurally and functionally diverse extracellular ligands to modulate organ development and function. NRP1 regulates tumor growth and vascular remodeling and modulates vascular permeability signaling pathways [34,35]. Our studies demonstrated that PDA regulates vascular endothelial barrier function through the activity of the VEGF signaling pathway and interaction with VE-Cad.

### 3.2. Diversity and Complexity of VE-Cad in Regulating Vascular Endothelial Barrier Function

VE-Cad is a critical endothelium-specific adhesion molecule and plays a pivotal role in maintaining and regulating endothelial barrier function [14,17,36]. Previous studies demonstrated interactions between VEGFR2 and VE-Cad, activation of VEGFR2 phosphorylation by VEGFa, interactions between VE-Cad and β-catenin and subsequent regulation of cell surface tension by the protein complex [37]. Our studies demonstrated that PDA significantly induced VE-Cad expression in endothelial cells, and enhanced VE-Cad expression caused vascular permeability characterized by macrophage infiltration. However, we treated endothelial cells with PDA and observed that VE-Cad was dynamically expressed in endothelial cells (Appendix A). Previous studies also demonstrated that soluble VE-cadherin is associated with and contributes to inflammation-induced breakdown of endothelial barrier functions [38]. Phosphorylation of VE-Cad is modulated by shear stress and is critical in regulating vascular permeability. Phosphorylated VE-Cad is internalized and ubiquitinated in response to increased vascular permeability [3]. Those studies demonstrated that diversity and complexity exist during the regulation of endothelial barrier dysfunction and, in addition to VE-Cad, it is likely that multiple signaling pathway are also involved in regulating endothelial barrier dysfunction.

### 3.3. Difference between Carotid Artery and Capillary Endothelial Cells

Partial ligation of the carotid arteries of ApoE knockout mice was used to determine whether PDA regulates vascular endothelial barrier integrity. We also observed that PDA induces NRP1 expression and activates the VEGF signaling pathway during angiogenesis following skeletal muscle injury. However, endothelial cells from different organs exhibit various morphologies and different biological behaviors. Our studies suggested that other signaling pathways are involved in PDA’s regulation of endothelial barrier integrity, and the NRP1/VEGFR2/VE-Cad signaling pathway controls several other biological processes.

These data demonstrate that PDA promotes endothelial barrier repair in pathological vascular remodeling at least partially through the NRP1/VEGFR2/VE-Cad signaling pathway.

## 4. Materials and Methods

### 4.1. Ethics Statement regarding Animal Experiments

The animals used in this study were approved by the Experimental Animal Ethics Committee of Chengdu University of Traditional Chinese Medicine (ethical approval number: 2019-04).

### 4.2. Partially Ligated Murine Carotid Artery model

ApoE knockout mice (ApoE^−/−^) aged 8–10 weeks were used in this study. Animals were fed a Western diet and received consecutive administrations of PDA (20 mg/kg) by intraperitoneal injection for 5 days. The carotid artery partial ligation was performed as previously described [29]. Briefly, anesthetized animals were given ketamine (80 mg/kg) and serazine (5 mg/kg), and the left carotid artery was separated and exposed under a microscope. The left external carotid, internal carotid and occipital artery were ligated, whereas the superior thyroid artery, which was the sole source for blood circulation, was untouched. Left carotid arteries were harvested at 7 or 28 days after consecutive PDA treatment.

### 4.3. Acute Skeletal Muscle Injury Model

Male ApoE^−/−^ mice aged 8–10 weeks were injected with PDA (20 mg/kg) intraperitoneally for 7 consecutive days. These mice were anesthetized with ketamine (80 mg/kg) and xylazine (5 mg/kg) by intraperitoneal injection. The skin on the tibialis anterior muscle was shaved and disinfected with iodine tincture. Then, 10 μM cardiotoxin working solution was prepared with sterilized phosphate-buffered saline, and 20 μL cardiotoxin working solution was intramuscularly injected into the tibialis anterior muscle. The anterior tibial muscles of animals treated with PDA were collected at day 5 and embedded in paraffin.

### 4.4. Hematoxylin and Eosin (H&E) Staining, Immunohistochemistry Staining (IHC) and Immunofluorescence Staining (IF)

The mouse carotid arteries were fixed with 4% paraformaldehyde overnight at 4 °C and embedded in paraffin, and 5 μm thick slides were collected. H&E staining was performed as previously described [27]. For IHC staining, antigen retrieval was performed with citric acid treatment at 98 °C for 5 to 10 min. The antigens were unmasked, incubated with primary antibody overnight at 4 °C and subjected to biotinylated secondary antibody incubation for 1 h (Vector Laboratories, Newark, CA, USA), following which ABC solution (Vector Laboratories) was incubated at room temperature for 30 min. The special targets were visualized after DAB solution was added. For IF staining, the deparaffinized slides were permeabilized with PBS containing 0.25% Triton-X-100, sealed with 10% goat serum, incubated with primary antibodies at 4 °C overnight and then incubated with Alexa-conjugated secondary antibody at room temperature for 1 h. Nuclei were visualized with 4′,6′-diamidino-2-phenylindole (DAPI) staining. For BRDU staining, DNA was denaturized using 2N HCl and incubated with antibodies. Images were taken using a confocal microscope (LS510, Zeiss, Oberkochen, Germany). The antibodies used in this study included CD31 (Cell Signaling Technology, Danvers, MA, USA), NRP1 (Proteintech, Chicago, IL, USA), VE-Cad (Cell Signaling Technology), VEGFR2 (Cell Signaling Technology), Mac2 (Sigma, St. Louis, MO, USA) and BRDU (Invitrogen, Waltham, MA, USA). 

### 4.5. Cell Culture

The human umbilical vein endothelial cell line and normal primary human umbilical vein endothelial cell were purchased from ATCC. The human umbilical vein endothelial cell line (HUVEC-C, ATCC CRL-1730) was cultured with F-12K medium (ATCC^®^ 30-2004™) containing 10% FBS. The normal primary human umbilical vein endothelial cell (HUVEC, ATCC PCS-100-010) was cultured with vascular cell basal medium (ATCC, PCS-100-030) supplemented with an endothelial Cell Growth Kit (ATCC, PCS-100-040).

### 4.6. HUVEC-C Cell Counting

A total of 5 × 10^4^ HUVECs (each well) were seeded in a six-well culture plate following treatment with PDA (10 μM). The cells were washed with PBS and trypsinized and the cell numbers were counted at 12 h, 24 h, 48 h and 72 h.

### 4.7. CCK8 Cell Proliferation Assay

A total of 1.5 × 10^3^ HUVEC-Cs (each well) were seeded in a 96-well culture plate and treated with PDA (10 μM) for 24 h. Absorbance at 450 nm was evaluated using a CCK8 kit.

### 4.8. BRDU Incorporation Assay

The BRDU incorporation assay was performed as previously described [39]. HUVEC-Cs were suspended in culture media containing PDA (10 μM), and then BRDU reagent labeling was undertaken for 24 h. Cells were washed in PBS and DNA was denaturized using 2N HCl for 30 min. After incubation, cells were fixed with 4% paraformaldehyde for 30 min at 4 °C and then washed in PBS. The fixed cells were immersed in citric acid solution (pH = 6) and heated in a water bath for 10 min. After washing in PBS, cells were incubated with 10% normal goat serum in PBS containing 0.3% Triton X-100 for 30 min at 4 °C. Cells were incubated overnight in anti-BRDU antibody in 10% normal goat serum containing 0.3% Triton X-100 at 4 °C. Cells were washed in PBS and incubated with secondary antibody in 10% goat serum and PBS with 0.3% Triton X-100 for 1 h at RT. IF staining was performed to identify the BRDU-incorporated cells. The images were taken using a confocal microscope (LS510, Zeiss).

### 4.9. Matrigel-Based Tube Formation Assay

HUVEC-Cs were treated with PDA (10 μM) overnight and seeded in growth factor-reduced Matrigel-coated (BD Bioscience, East Rutherford, NJ, USA) 24-well plates (8 × 10^4^ cells in each well) for 48 h. After calcein AM staining, the images of tubes were taken with an inverted immunofluorescence microscope. Cumulative tube numbers and tube lengths were quantified with Image J software 1.53t (National Institutes of Health, Wayne Rasband, Bethesda, MD, USA).

### 4.10. Propidium Iodide (PI) Flow Cytometric Assay

HUVEC-Cs were treated with PDA (10 μM), trypsinized and fixed with 70% ethanol. The cell cycle was analyzed using flow cytometry after propidium iodide staining (BD Biosciences).

### 4.11. Boyden Chamber Migration Assay

A total of 1 × 10^6^ HUVEC-Cs were suspended in 100 uL FBS-free culture media and seeded in a Boyden chamber (FALCON). The Boyden chamber was set up with a 24-well culture plate that contained 500 uL complete culture medium (10% FBS) and 10 μM PDA. Cell numbers were counted after crystal violet staining

### 4.12. Spheroid Sprouting Assay

The spheroid sprouting assay was performed as described previous [27]. Methylcellulose solution was prepared by dissolving 6 g methylcellulose (sigma) in 250 mL of prewarmed serum-free medium, and 250 mL of DMEM containing 10% serum was added. Suspended cells were dissolved in methylcellulose solution prepared with 10 mL methylcellulose solution and 40 mL culture medium to form the spheres. Neutralized collagen solution was added to a 24-well culture plate and incubated at 37 °C until collagen solidified. Spheres were mixed with dissolved collagen solution and transferred to a collagen-solidified culture plate. The culture plate was allowed to solidify for 30 min at 37 °C, 200 uL of complete medium containing PDA was added and the mixture was cultured overnight. The spheroid sprouting was visualized after calcein AM staining. Images were captured using a confocal microscope (Leica Microsystem CMS GmbH, Wetzlar, Germany), and the sprouting numbers and sprouting lengths of each sphere were analyzed with Image J software 1.53t (National Institutes of Health, Wayne Rasband, Bethesda, MD, USA).

### 4.13. Quantitative Real-Time PCR Analysis

Total RNA was extracted from HUVEC-Cs with Trizol reagent. Quantification of RNA was undertaken using a spectrophotometer (Denovix, Wilmington, DE, USA). Using 600 ng RNA as a template, the cDNA were amplified with an iScrip cDNA synthesis kit after reverse transcription with random hexamer primers. Real-time PCR was performed in duplicate for each sample on a Bio-Rad real-time PCR system. The primer sequences used in this study are provided in the Appendix A. The relative gene expression level was analyzed with the 2^−ΔΔct^ method.

### 4.14. Protein Extraction and Western Blotting

Proteins from HUVECs and HUVEC-Cs were extracted using RIPA lysis buffer. Protein concentration was evaluated with a BCA kit (Biosharp). Proteins were denatured at 98 °C for 10 min, separated using sodium dodecyl sulfate-polyacrylamide gel electrophoresis (SDS-PAGE) and transferred onto polyvinylidene fluoride (PVDF) membranes. After blockage with 5% fat skim milk, they were incubated with specific antibodies at 4 °C overnight. Images were captured using an ImageQuan LAS4000 Image Station and the densities of protein bands were quantified using Imagequant TL software 8.1 (GE Healthcare, Chicago, IL, USA).

### 4.15. Molecular Docking Simulation of PDA with NRP1, VE-Cad and VEGFR2

A molecular docking simulation was performed to determine the binding energy of PDA with NRP1, VE-Cad and VEGFR2 using Autodock Vina 1.5.6 software, which was developed by Olson’s research group [40]. For binding energy values less than zero, those proteins were considered to spontaneously bind and interact with each other.

### 4.16. CO-Immunoprecipitation Assay

Total proteins were extracted from HUVECs using RIPA buffer. Cell lysate was precleared using anti-species-specific IgG beads. Precleared cell lysate was incubated with NRP1 (Proteintech), VE-Cad (Cell Signaling Technology) and VEGFR2 (Cell Signaling Technology) for 1 h at 4 °C. Then, the lysate was incubated with pre-equilibrated protein A/G agarose beads on a rocking platform overnight at 4 °C. The CO-immunoprecipitated targets were evaluated with Western blotting.

### 4.17. THP-1 Adhesion Assay

HUVEC-Cs were treated with PDA (10 μM) and seeded in growth factor-reduced Matrigel-coated (BD Bioscience) six-well plates (2 × 10^5^ cells in each well) for 9 h. Then, 4 × 10^4^ THP-1 monocytes were suspended in 1 mL of 1640 complete medium and seeded in a HUVEC-C culture plate. THP-1 adhesion was monitored and images were captured using an inverted microscope after 1 h incubation.

### 4.18. siRNA Transfection

Scrambled siRNA and siRNA targeting NRP1 were synthesized by GenePharma. The siRNAs were transfected into HUVECs using Lipofectamin 2000 reagent following the manufacturer’s protocol. The sequence used for siRNA transfection is shown in Appendix A.

Quantitative data are presented as means ± SEM. Statistical analysis was performed in GraphPad prism software 8.0.1.244 (GraphPad Software, Inc., San Diego, CA, USA). Normal distribution was evaluated with the Kolmogorov–Smirnov test, and statistical comparisons between the two groups were undertaken using two-tailed unpaired Student’s *t*-tests or one- or two-way analysis of variance (ANOVA), followed by Bonferroni’s post hoc tests when appropriate. Two-sided *p* values were also quantified. * *p* < 0.05 was statistically significant.

## Figures and Tables

**Figure 1 ijms-24-03096-f001:**
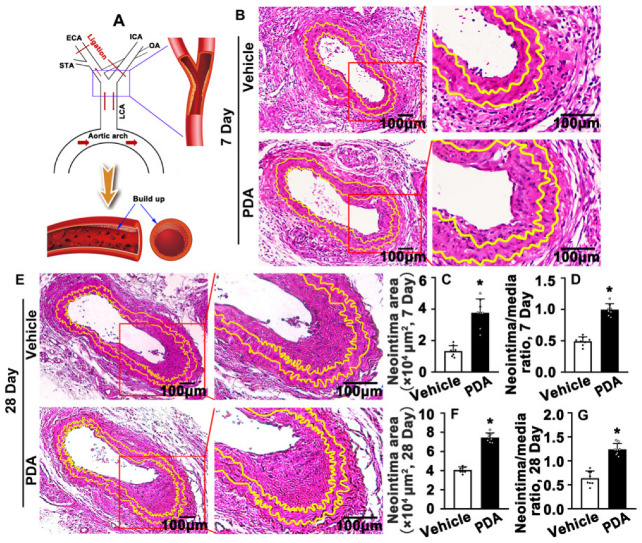
PDA promotes partial carotid artery ligation-induced neointima formation in a model of vascular injury. (**A**) Schematic diagram of the left carotid artery partial ligation model showing a ligation injury caused by vascular remodeling. (**B**) Partial ligation of the left carotid artery in ApoE^−/−^ mice. PDA (20 mg/kg) was administered by intraperitoneal injection for 7 consecutive days. The arteries were harvested and embedded in paraffin. A 5 μm thick paraffin section was collected at the ligation site. H&E staining was performed to observe the morphological changes in blood vessels. (**C**,**D**) Analysis areas of the neointimal hyperplasia and ratio of the neointimal areas to the medium layer area (*n* = 8). (**E**) Partial ligation of the left carotid artery in ApoE^−/−^ mice. PDA (20 mg/kg) was administered by intraperitoneal injection for 21 consecutive days. Representative image of H&E staining. (**F**,**G**) Analysis areas of the neointimal hyperplasia and ratio of the neointimal areas to the medium layer area (*n* = 8). Data are expressed as means ± SEM. * *p* < 0.05.

**Figure 2 ijms-24-03096-f002:**
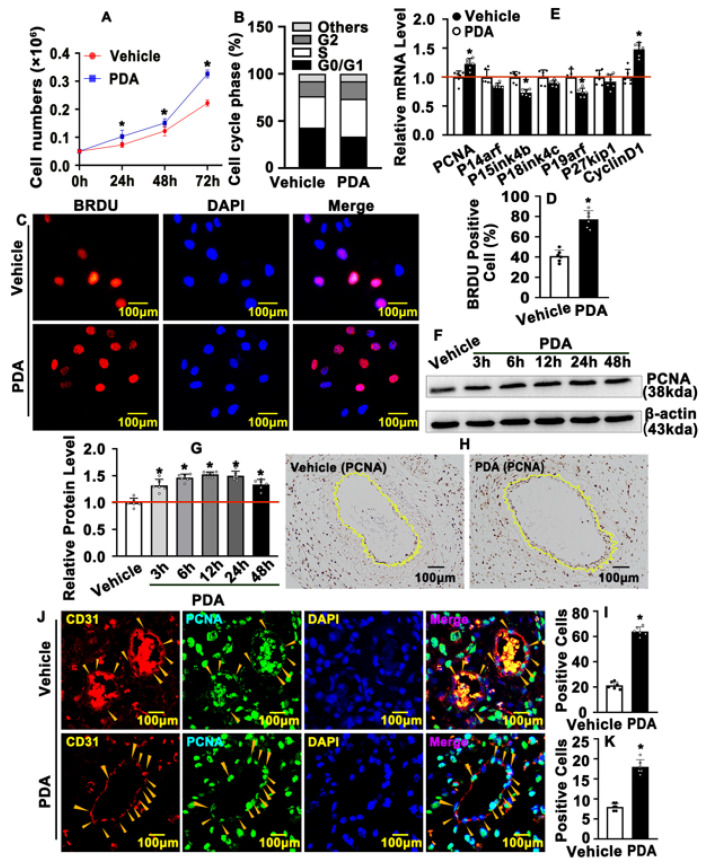
PDA promotes vascular endothelial cell proliferation. (**A**) HUVECs were treated with PDA (10 μM), the cell numbers were counted at different time points (24 h, 48 h, 72 h) (*n* = 6). (**B**) HUVEC-Cs were treated with PDA (10 μM, 24 h) and the cell cycle was determined using flow cytometer analysis after propidium iodide (PI) staining. (**C**) HUVEC-Cs were treated with BRDU-labeling buffer for 20 h following PDA (10 μM) treatment for 24 h. IF staining was performed to determine BRDU-incorporated HUVEC-Cs. (**D**) The BRDU-positive cells were quantitatively evaluated (*n* = 6). (**E**) HUVEC-Cs were treated with PDA (10 μM, 24 h), and the mRNA levels of proliferation-related genes were detected using real-time PCR (*n* = 6). (**F**) Western blotting was used to detect the expression of PCNA in HUVEC-Cs at different time points after PDA (10 μM) treatment. The quantitative data are shown in (**G**) (*n* = 8). (**H**) IHC staining was performed against the proliferation marker gene PCNA on the left carotid artery partial ligation model following PDA treatment for 7 days. PCNA-positive cells in neointimal areas are shown in (**I**) (*n* = 8). (**J**) IF staining was used to evaluate the expression of PCNA in blood vessels within the anterior tibial muscle of mice that were intramuscularly injected with cardiotoxin and treated with PDA (20 mg/kg) for 5 consecutive days. Yellow arrows indicate regions containing CD31–positive and PCNA-positive vascular endothelial cells. PCNA-positive cells in neointimal areas are shown in (**K**) (*n* = 8). Analytical data are expressed as means ± SEM, * *p* < 0.05.

**Figure 3 ijms-24-03096-f003:**
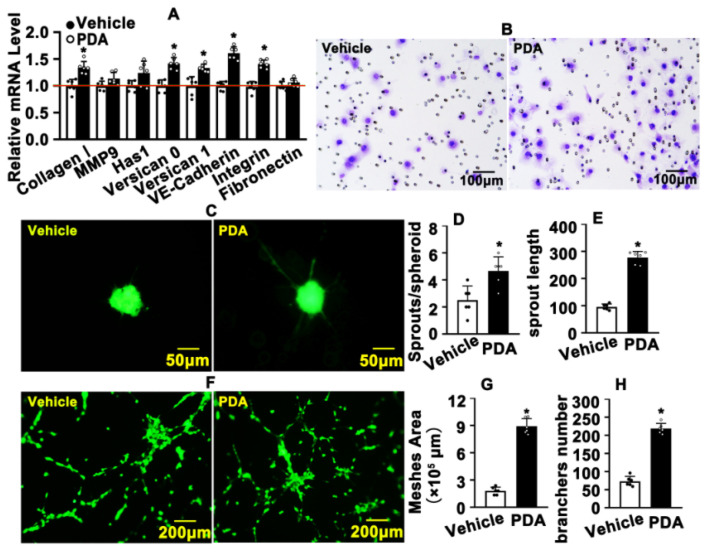
PDA promotes vascular endothelial cell migration. (**A**) HUVEC-Cs were treated with PDA (10 μM) for 24 h and real-time PCR was performed to evaluate transcription levels of migrative-related genes (*n* = 6). (**B**) The Boyden chamber cell migration assay was performed following PDA (10 μM, 24 h) treatment, and the migrated cells were visualized with crystal violet staining. (**C**) A spheroid sprouting assay was performed in the presence of PDA (10 μM, 24 h), and the sprouting of HUVEC-Cs was visualized using calcein AM staining. Quantification of sprouts and total sprout lengths are exhibited in (**D**,**E**) (*n* = 6). (**F**) The Matrigel-based tube formation assay was performed after PDA (10 μM) treatment for 48 h. The branch numbers and mesh areas are quantified in (**G**,**H**) (*n* = 6). The analysis data are expressed as means ± SEM. * *p* < 0.05.

**Figure 4 ijms-24-03096-f004:**
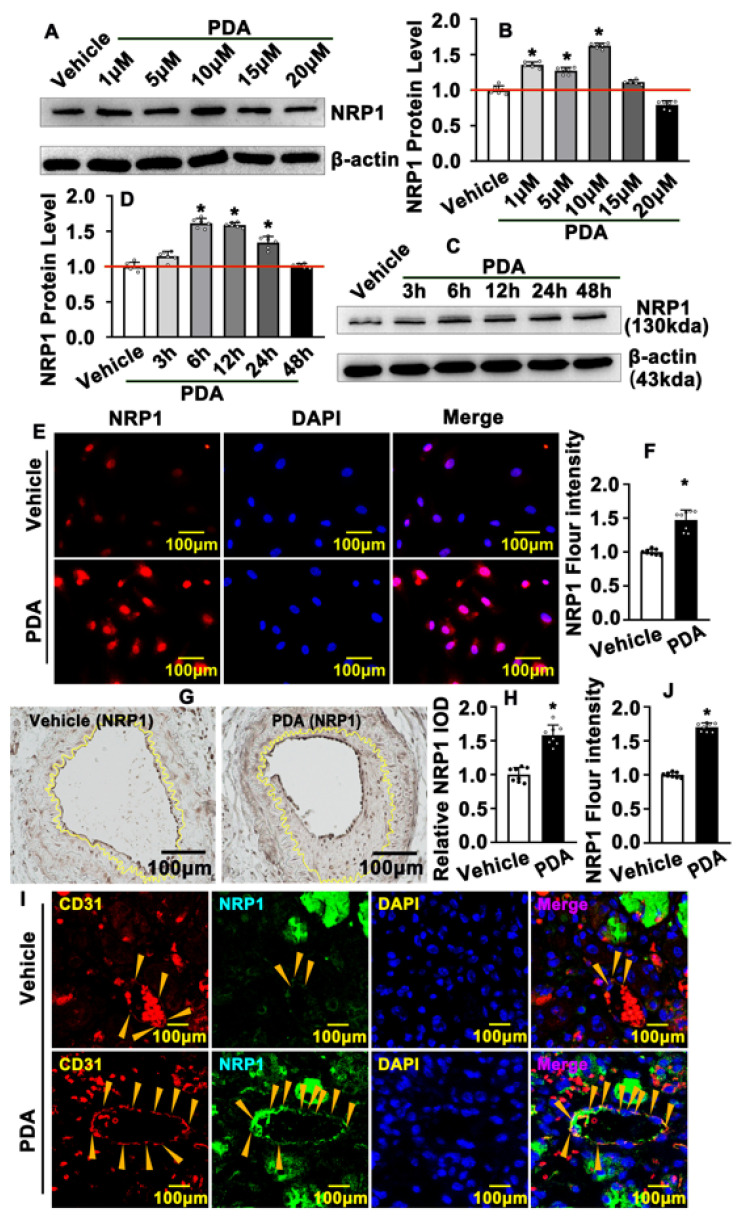
PDA induces vascular endothelial cell NRP1 expression after vascular injury. (**A**) Western blotting was performed to determine the expression of NRP1 in HUVEC-Cs after PDA (10 μM, 24 h) treatment with different doses, and the quantification data are shown in (**B**) (*n* = 8). (**C**) Western blotting was performed to determine the expression of NRP1 in HUVEC-Cs after PDA (10 μM) treatment at different time points, and the quantification data are evaluated in (**D**) (*n* = 8). (**E**) IF staining was performed to detect the expression of NRP1 in the HUVECs after PDA (10 μM, 24 h) treatment, and the quantification data are evaluated in (**F**) (*n* = 8). (**G**) IHC staining was performed against NRP1 antibody in a left carotid artery partial ligation model following PDA treatment for 28 days, and the quantification data are evaluated in (**H**) (*n* = 8). (**I**) IF staining was used to evaluate the expression of NRP1 in blood vessels within the anterior tibial muscle of mice that were intramuscularly injected with cardiotoxin and treated with PDA (20 mg/kg) for 5 consecutive days. Yellow arrows indicate regions containing CD31–positive and NRP1-positive vascular endothelial cells.The quantification data are evaluated in (**J**) (*n* = 8). Analytical data are expressed as means ± SEM * *p* < 0.05.

**Figure 5 ijms-24-03096-f005:**
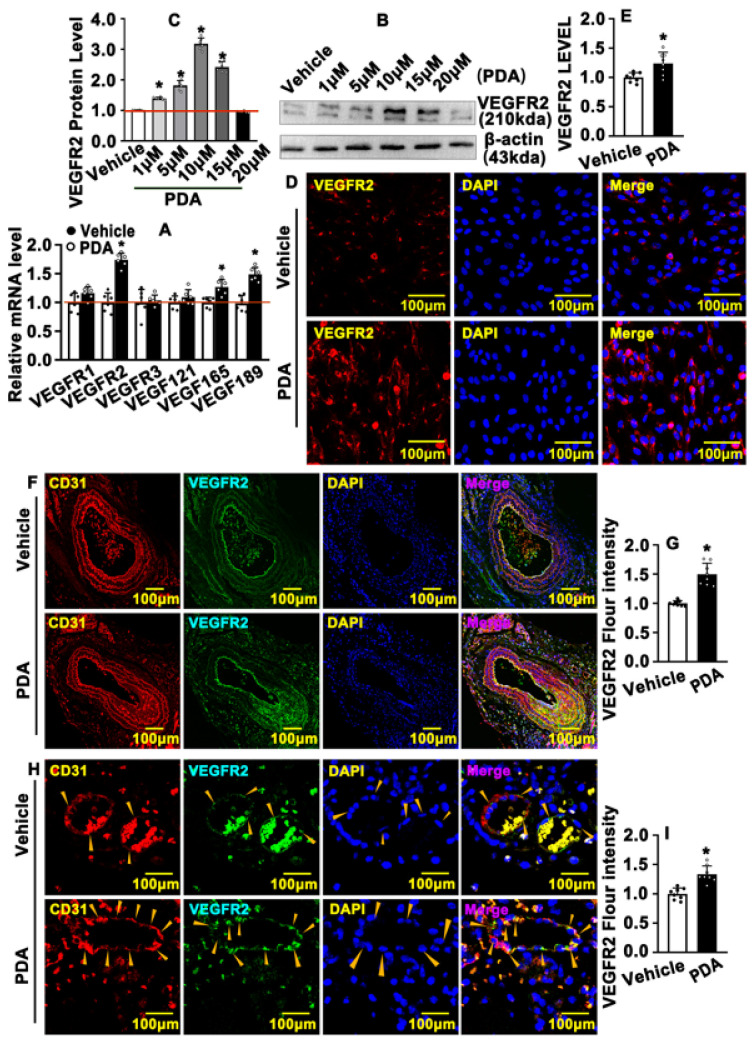
PDA activates the VEGF signaling pathway after vascular injury. (**A**) HUVEC-Cs were treated with PDA (10 μM) for 24 h. The transcription levels of the VEGF family were evaluated with real-time PCR (*n* = 6). (**B**) Western blotting was performed to determine the expression of VEGFR2 in HUVECs after PDA (10 μM, 24 h) treatment at different doses, and the quantification data are evaluated in (**C**) (*n* = 8). (**D**) IF staining was performed to detect the expression of VEGFR2 in the HUVECs after PDA (10 μM) treatment for 24 h, and the quantification data are evaluated in (**E**) (*n* = 8). (**F**) IHC staining was performed against CD31 and VEGFR2 antibodies to evaluate the expression of VEGFR2 in a left carotid artery partial ligation model following PDA treatment for 28 days. Yellow arrows indicate regions containing CD31–positive and VEGFR2-positive vascular endothelial cells. The quantification data are evaluated in (**G**) (*n* = 8). (**H**) IF staining was used to evaluate the expression of VEGFR2 in blood vessels within the anterior tibial muscle of mice that were intramuscularly injected with cardiotoxin and treated with PDA (20 mg/kg) for 7 consecutive days, and the quantification data are evaluated in (**I**) (*n* = 8). Analytical data are expressed as means ± SEM * *p* < 0.05.

**Figure 6 ijms-24-03096-f006:**
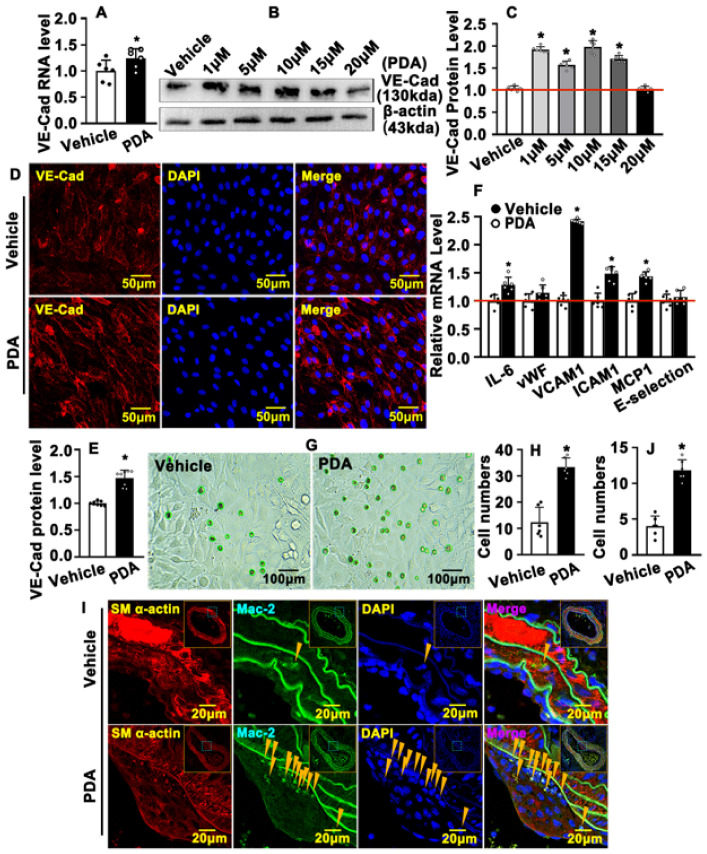
PDA causes VE-Cad-dependent vascular endothelial barrier dysfunction. (**A**) HUVEC-Cs were treated with PDA (10 μM) for 12 h. Real-time PCR was performed to detect the mRNA levels of VE-Cad (*n* = 6). (**B**) Western blotting was performed to determine the expression of VE-cadherin in HUVEC-Cs after PDA (10 μM, 24 h) treatment at different doses, and the quantification data are evaluated in (**C**) (*n* = 8). (**D**) IF staining of VE-Cad antibody was performed to evaluate the expression of VE-Cad in HUVECs after PDA treatment for 24 h, and the quantification data are evaluated in (**E**) (*n* = 8). (**F**) HUVEC-Cs were treated with PDA (10 μM) for 24 h. Real-time PCR was performed to detect the mRNA levels of the inflammation-related genes (*n* = 6). (**G**) HUVEC-C and THP-1 were co-cultured and treated with PDA (10 μM) for 12 h to evaluate their relationship and the changes in the process of vascular remodeling, and the quantification data are evaluated in (**H**). (**I**) IHC staining was performed against SM a-actin and Mac-2 antibodies to evaluate the level of macrophage infiltration in a left carotid artery partial ligation model following PDA treatment for 7 days. Yellow arrows indicate regions containing Mac-2–positive macrophages. The quantification data exhibited in (**J**) (*n* = 8). Analytical data are expressed as means ± SEM * *p* < 0.05.

**Figure 7 ijms-24-03096-f007:**
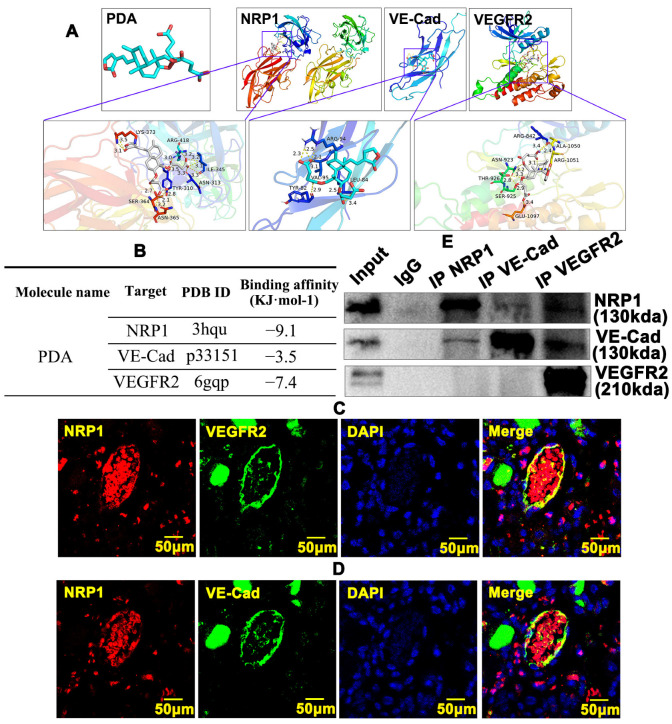
PDA regulates interactions between NRP1, VEGFR and VE-Cad within vascular endothelial cells. (**A**) Molecular docking simulation performed to evaluate the binding energy of PDA with NRP1, VEGFR2 and VE-Cad using Autodock Vina 1.5.6 software, which was developed by Olson’s research group. PDA worked as a receptor, and NRP1, VEGFR2 and VE-Cad were used as ligands to detect the docking sites between receptors and ligands. The three-dimensional structures of VEGFR2 and NRP1 were obtained from the RCSBPDB database (http://www.rcsb.org/, accessed on 16 August 2022) and that of VE-Cad was obtained from the SWISS-Model database (http://swissmodel.expasy.org/, accessed on 20 August 2022). When values for the binding energy were less than zero, those proteins were considered to spontaneously bind and interact with each other. (**B**) The binding energies of PDA with NRP1, VEGFR2 and VE-Cad were determined based on a molecular docking simulation. (**C**) IF staining of NRP1 and VEGFR2 was performed in the skeletal muscle injury model to determine their co-localization expression in vascular endothelial cells. (**D**) Immunofluorescence staining of NRP1 and VE-Cad was performed in the skeletal muscle injury model to determine their co-localization expression in vascular endothelial cells. (**E**) The interaction between NRP1, VEGFR2 and VE-Cad was validated using the CO-immunoprecipitation assay in HUVECs.

**Figure 8 ijms-24-03096-f008:**
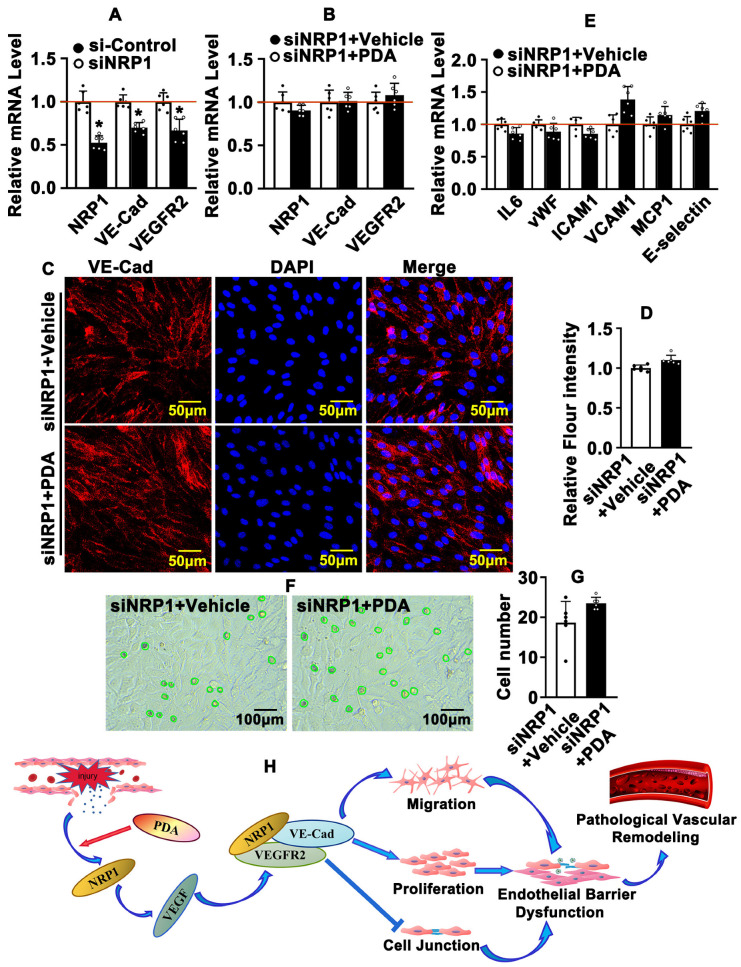
PDA regulates vascular endothelial barrier function through the NRP1/VEGFR2/VE-Cad signaling pathway. (**A**) After knockdown of NRP1 in HUVECs, real-time PCR was performed to evaluate transcription levels of NRP1-, VEGFR2- and VE-Cad-related genes (*n* = 6). (**B**) After knockdown of NRP1, HUVECs were treated with PDA (10 μM) for 24 h and real-time PCR was performed to evaluate transcription levels of NRP1-, VEGFR2- and VE-Cad-related genes (*n* = 6). (**C**) After knockdown of NRP1, IF staining was performed to detect the expression of VE-Cad in the HUVECs following PDA (10 μM) treatment for 24 h, and the quantification data are evaluated in (**D**) (*n* = 8). (**E**) After knockdown of NRP1, HUVECs were treated with PDA (10 μM) for 24 h and real-time PCR was performed to evaluate transcription levels of inflammation-related genes (*n* = 6). (**F**) After knockdown of NRP1, HUVEC-C and THP-1 were co-cultured and treated with PDA (10 μM) for 12 h, and the quantification data are evaluated in (**G**). (**H**) Schematic diagram indicating that PDA promoted endothelial barrier repair in pathological vascular remodeling. Analytical data are expressed as means ± SEM * *p* < 0.05.

## Data Availability

Not applicable.

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
