# Peer review of "Potassium Dehydroandrograpolide Succinate Targets NRP1 Mediated VEGFR2/VE-Cadherin Signaling Pathway to Promote Endothelial Barrier Repair"

_ijms, 2023, doi:10.3390/ijms24043096_

Round 1

Reviewer 1 Report

The present paper by Wang et al examined the role of NRP1 in mediating VEGFR2/VE-cadherin signaling in response to potassium dehdryandrograpolide succinate (PDA), a derivative of andrographolide. The findings presented suggest that PDA promotes vascular dysfunction i.e., intimal hyperplasia, cell proliferation, etc.) both alone and in a model of vascular injury.

Comments to be Addressed:

1.         “Andrographolide” and “Potassium Dehdroandrograpolide Succinate” do not need to be capitalized when appearing in the middle of a sentence.

2.         What are the pharmokinetics of injection of PDA (20 mg/kg, ip)? Please included some details of absorption and clearance of PDA in the first pass circulation before it reaches endothelium of the carotid artery.

3.         The relative effects of in vivo PDA administration appear to be larger in terms of hyperplasia than the in vitro effect of PDA. Are these effects simply concentration or dose-dependent (20 mg/kg in vivo versus 10 µM in vivo)?

4.         How were the single doses of PDA determined? Are any of the effects in any of the assays in the present study dose-dependent? Some of the latter experiments somewhat address this (NRP1 expression), however the effects of PDA appear biphasic (i.e. NRP1 expression).

5.         It is not clear how PDA will be clinically useful in the context of vascular injury? Please include some discuss on the benefit of PDA.

6.         Page 1, title: The title is awkward and confusing as written, please revise. Also, please avoid the use of abbreviations in the title.

7.         Page 1, abstract: “For underly mechanism studies”?

8.         Page 1, abstract: “Our study demonstrated that PDA plays a critical role in regulating endothelial barrier impairment induced pathological vascular remodeling.”?? - Not sure what this sentence is saying. It appears that PDA enhances vascular impairment, no?

9.         Page 2, line 45: “Endothelial” does not need to be capitalized.

10.       Page 2, line 46: Change “functions” to “function”

11.       Page 2, lines 75-76: The single sentence paragraph should be re-worded to properly reflect the aim of the study, inclusion of a hypothesis would also be helpful here.

12.       Figure 1, Legend: As stated the first sentence is confusing - “PDA promotes partially carotid artery ligation induced neointima formation.” This confusion could be reduced by removing excess verbiage, eg “PDA promotes partially carotid artery ligation induced neointima formation in a model of vascular injury.”

13.       Page 16, lines 376-379: This sentence is awkward and confusing as written. 

14.       Please include indices of molecular weight associated with Western Blot images.

Reviewer 2 Report

This work deals with determination and monitoring of the regulation of pathological vascular remodelling by potassium hydroandrograpolide succinate via a modulating integrity of endothelial barrier.

Different approaches such as flow cytometry, BRDU-based, Boyden chamber cell migration, Spheroid sprouting and Matrigel-based assay were used for such a purpose.

The ultimate overcome was reported  as: PDA confirmed to induce pathological vascular remodelling characterised by enhanced neointima formation.

REMARKS

1 In section 1. “Introduction”, the last paragraph, provide more information about the outcomes of the suggested work.

2 In 4.3 section, “BRDU incorporation assay”, follow the upgrade provided here proper citing of the another BRDU work.

For BRDU staining [https://doi.org/10.1016/j.vascn.2015.05.012], DNA denaturized using 2N HCl, and following antibodies incubation. The image was taken using a confocal microscopy (LS510, Zeiss). Antibodies used in this study included CD31 (Cell Signaling Technology), NRP1 (Proteintech), VE-Cad (Cell Signaling Technology), VEGFR2 (Cell Signaling Technology), Mac2 (Sigma), BRDU (Invitrogen). The image was taken using a confocal microscopy (LS510, Zeiss).

3 I miss a “Conclusion” section to incorporate and sum up the ultimate outcomes and future aims of the authors in this scope of investigation. Please, made one.

Author Response

Point 1: In section 1. “Introduction”, the last paragraph, provide more information about the outcomes of the suggested work.

Response 1: Thank you for your comment. We have revised the last paragraph of the discussion section so that the purpose of our study can be more clearly expressed. Please see page 2, line 75-83 for specific details.

Point 2: In 4.3 section, “BRDU incorporation assay”, follow the upgrade provided here proper citing of the another BRDU work.

“For BRDU staining [https://doi.org/10.1016/j.vascn.2015.05.012], DNA denaturized using 2N HCl, and following antibodies incubation. The image was taken using a confocal microscopy (LS510, Zeiss). Antibodies used in this study included CD31 (Cell Signaling Technology), NRP1 (Proteintech), VE-Cad (Cell Signaling Technology), VEGFR2 (Cell Signaling Technology), Mac2 (Sigma), BRDU (Invitrogen). The image was taken using a confocal microscopy (LS510, Zeiss).

Response 2: Thank you for your suggestion. We have revised the methodological description of the BRDU incorporation assay and enhanced it with the relevant literature provided by the reviewer and added it to the references. Please see section 4.8 for specific details.

Point 3: I miss a “Conclusion” section to incorporate and sum up the ultimate outcomes and future aims of the authors in this scope of investigation. Please, made one.

Response 3: Thanks a lot for your comment. The conclusion of this study is “PDA plays a critical role in promoting endothelial barrier repair in pathological vascular remodeling”. We revised the section 3 “Discussion” to make a more comprehensive and clear summary of the results and conclusions for this study. Reasonable hypotheses were made to clarify the direction and purpose of future research for the problems that remain to be solved in the experiment, so that we can improve the shortcomings in this study. Please see page17, line 384-401 for specific details.

Round 2

Reviewer 2 Report

Authors have reacted to given queries